materials science

ferroelectric, KNN, epitaxial, aqueous chemical solution deposition

**Author for correspondence:**
Mari-Ann Einarsrud
e-mail: mari-ann.einarsrud@ntnu.no

This article has been edited by the Royal Society of Chemistry, including the commissioning, peer review process and editorial aspects up to the point of acceptance.

# Epitaxial $K_{0.5}Na_{0.5}NbO_3$ thin films by aqueous chemical solution deposition

Ky-Nam Pham[1], Nikolai Helth Gaukås[1],
Maxim Morozov[1,4], Thomas Tybell[2], Per Erik Vullum[3,5],
Tor Grande[1] and Mari-Ann Einarsrud[1]

[1]Department of Materials Science and Engineering, [2]Department of Electronic Systems, and [3]Department of Physics, NTNU Norwegian University of Science and Technology, Trondheim, Norway
[4]Laboratory of Solution Chemistry of Advanced Materials and Technologies, ITMO University, St. Petersburg, Russian Federation
[5]SINTEF Industry, 7465 Trondheim, Norway

M-AE, 0000-0002-3017-1156

We report on an environmentally friendly and versatile aqueous chemical solution deposition route to epitaxial $K_{0.5}Na_{0.5}NbO_3$ (KNN) thin films. The route is based on the spin coating of an aqueous solution of soluble precursors on $SrTiO_3$ single crystal substrates followed by pyrolysis at 400°C and annealing at 800°C using rapid thermal processing. Strongly textured films with homogeneous thickness were obtained on three different crystallographic orientations of $SrTiO_3$. Epitaxial films were obtained on (111) $SrTiO_3$ substrates, while films consisting of an epitaxial layer close to the substrate followed by an oriented polycrystalline layer were obtained on (100) and (110) $SrTiO_3$ substrates. A $K_2Nb_4O_{11}$ secondary phase was observed on the surface of the thin films due to the evaporation of alkali species, while the use of an NaCl/KCl flux reduced the amount of the secondary phase. Ferroelectric behaviour of the films was investigated by PFM, and almost no dependence on the film crystallographic orientation was observed. The permittivity and loss tangent of the films with the NaCl/KCl flux were 870 and 0.04 (100-orientation) and 2250 and 0.025 (110-orientation), respectively, at 1 kHz.

## 1. Introduction

Lead-free piezo- and ferroelectric materials have received considerable attention owing to the environmental issues with the state-of-the-art lead zirconate titanate (PZT)-based materials.

A number of lead-free piezoelectric polycrystalline materials have been developed and, among them, potassium sodium niobate $K_xNa_{1-x}NbO_3$ (KNN) shows a potential for the replacement of the high performance PZT-based materials, especially if the piezoelectric performance is enhanced by texturing [1–4]. Moreover, the use of highly oriented thin films of KNN-based compositions is promising for microelectromechanical applications [3,5]. KNN-based thin films have been grown by various physical methods such as pulsed laser deposition [6], radio frequency magnetron sputtering [7], metal-organic chemical vapour deposition [8] and atomic layer deposition [9]. However, to bring these films to the market, chemical solution deposition routes have to be developed as this deposition technique is industrially upscalable and is the technology largely used for the PZT-based films. Chemical solution deposition of oxide thin films is simple and versatile, and several groups have proposed various routes for KNN-based materials; however, there is a challenge with the formation of secondary phases.

The first chemical solution routes developed for KNN-based thin films are mostly based on alkali acetates and niobium alkoxide as precursors and the use of 2-methoxyethanol as solvent [5,10–14]. Excess alkali has been found to be necessary to reduce the formation of niobium-rich secondary phases. Ahn et al. reported that KNN thin films deposited on Pt substrate with 20 mol% alkali excess, lowered the leakage current density and the films exhibited saturated $P–E$ hysteresis loops [10] due to the reduction of the amount of the $K_4Nb_6O_{17}$ secondary phase. Kang et al. [12] studied the effect of annealing procedure using a similar route based on alkoxides and obtained (100) oriented films on platinized Si using a layer-by-layer annealing method. Vendrell et al. [13] deposited films using a similar recipe on (100) $SrTiO_3$ (STO) and obtained highly textured films with out-of-plane orientation in the [001] direction. However, because 2-methoxyethanol is poisonous and harmful to the environment, there has been a search for alternative routes. Kwak et al. [15] replaced 2-methoxyethanol with the environmentally more compatible solvent ethanol in the acetate/alkoxide-based sol–gel route and developed KNN films with dense and uniform microstructure. The maximum dielectric constant of the films was 2000 at 10 Hz, showing the potential of this method. To avoid the use of alkoxides, Yao et al. [16] used a precursor based on $Nb_2O_5$ and deposited films on Ti substrates. The films prepared by the new precursor showed comparable dielectric constant of 480 at 100 Hz to films prepared by using Nb-alkoxide precursor. Another attempt to use aqueous processing of KNN thin films was reported by Zhang et al. [17] where water was used in a combination with PVA and an Nb-citrate complex as a precursor. The introduction of PVP was claimed to induce (100) oriented growth on platinized Si giving a dielectric constant of 280 at 1 kHz.

Highly textured KNN-based thin films have been obtained both on $Pt/TiO_2/SiO_2/Si$ and single crystal STO substrates using chemical solution deposition [5]. However, stoichiometry control is challenging as alkali volatility is shown to induce the formation of niobium-rich secondary phases. The volatility of alkali has been carefully studied by Wang et al. [18] using a combination of TGA and mass spectroscopy. They prepared the films from a standard alkoxide-based route in 2-methoxyethanol, using the organic additives ethylenediaminetetraacetic acid (EDTA), diethanolamine (DEA) and ethanolamine (MEA) which decreased the crystallization temperature. Volatility of potassium and sodium species were claimed to occur at temperatures as low as $200°C$, hence explaining the formation of the niobium-rich secondary phases. The addition of a combination of EDTA + DEA + MEA reduced the volatility of alkali by 80–90%. The reduced volatility of alkali suppressed the nucleation rate, and films with large grains were formed [18].

Here we report on a simple, aqueous-based environmentally friendly route to chemical solution deposition of $K_{0.5}Na_{0.5}NbO_3$ thin films on STO substrates. The preparation route is based on an aqueous solution of precursor salts deposited by spin coating onto the substrate. The effect of the orientation of the substrate on the thin film crystallization, microstructure and piezoelectric properties is discussed. The formation of a niobium-rich secondary phase at the surface of the films is significantly reduced by introducing a salt flux during the film crystallization. Finally, the formation of secondary phases in this system due to alkali volatility is discussed.

# 2. Experimental methods

Starting chemicals for the KNN precursor solutions were $NaNO_3$ (Sigma-Aldrich, St. Louis, USA), $KNO_3$ (Merck, Darmstadt, Germany) and $(NH_4)NbO(C_2O_4)_2·5H_2O$ (H. C. Starck, Goslar, Germany). $KNO_3$ and $NaNO_3$ were pre-dried at $250°C$ prior to use. The Nb-ammonium oxalate was first dissolved in distilled water at $80°C$ for 16 h. The concentration of the niobium solution was adjusted to approximately 0.20 M

and the concentration was determined by thermogravimetric analysis. The solution was further mixed with 25 mol% excess of $KNO_3$ and $NaNO_3$ to compensate for alkali loss during processing. The solution was stirred at 70°C for 5 h to obtain approximately 0.2 M Nb−K−Na aqueous precursor solution. In addition, a 1 : 1 molar ratio mixture of NaCl/KCl salts (Sigma-Aldrich, St. Louis, US) (total 25 mol% relative to KNN) was added to selected precursor solutions by dissolving the chlorides in the solution. The NaCl/KCl mixture was proposed to act as a flux during the crystallization of the films. In order to increase the wettability of the precursor solutions on the substrate, 1 wt% of a solution of 15 wt% polyvinyl alcohol (MW 40−88, Merck, Darmstadt, Germany) in distilled water was added as a surfactant.

The stable precursor solutions were deposited onto (100), (110) and (111) oriented $SrTiO_3$/Nb:$SrTiO_3$ (STO/Nb:STO) substrates (Crystal GmbH, Berlin, Germany) by spin coating at 2500−3000 r.p.m. for 20−40 s. The films were dried at 200°C for 2 min using a hotplate followed by pyrolysis at 400°C for 5 min by rapid thermal processing (RTP) with a heating rate of 40 K s$^{-1}$ (Jipelec Jetfirst 200 mm, Semco Technologies, Montpellier, France). In order to increase the film thickness, the deposition, drying and pyrolysis steps were repeated 15 times. The final films were heat treated at 800°C for 5 min in flowing oxygen by RTP with a heating rate of 40 K s$^{-1}$. The films with excess alkali are named KNN-E-xxx and the ones with the addition of the salt flux in addition to excess alkali are named KNN-ES-xxx where xxx describes the orientation of the substrate (100, 110 or 111).

Phase purity of the thin films was determined by X-ray diffraction (XRD) (Siemens D5005, Germany) analysis using CuK$\alpha$. Regular $\theta - 2\theta$ scans were performed in continuous mode in the $2\theta$ range from 20 to 80° with a step size of 0.01° and dwell time of 2 s at each step. Grazing incidence XRD was performed both on Siemens D5005 and Bruker D8 Discover diffractometers by using a grazing incidence angle of 3° in the $2\theta$ range from 10 to 80°. A scanning step of 0.08° and a counting time of 16 s per step were used.

Cross-section transmission electron microscopy (TEM) samples of selected films were prepared by focused ion-beam milling (FIB) with a dual-beam Helios Nanolab 600 from FEI. The final polishing with the Ga ion-beam was done at 5 kV. The entire sample was coated with 30 nm of Au prior to the FIB preparation to avoid sample charging. Electron beam assisted Pt deposition was further used to coat the region chosen for TEM to avoid ion-beam damage of the KNN film. Secondary electron SEM images of the film cross-section were recorded in the FIB. A Cs probe- and image-corrected cold-FEG JEOL ARM 200F, operated at 200 kV, was used to characterize the sample. The TEM was equipped with a Centurio silicon drift detector (0.98 sr solid angle) for EDX and a Quantum GIF with dual electron energy loss spectroscopy (EELS).

The electronic properties were studied by piezoresponse force microscopy (PFM), a scanning probe microscopy technique, and impedance analysis. The local piezoelectric properties were studied by PFM applying an AC bias between the tip and the sample bottom electrode (Nb:STO), resulting in a local piezoresponse (PR). PR imaging was carried out using a MultiMode series AFM with a Digital Instruments Nanoscope IIIA controller equipped with conductive tips (Pt/Ir-coated silicon) to serve as a top electrode. The PR signal is composed of a phase and an amplitude (PR amplitude and PR phase). Local piezoelectric loops were obtained by subsequently adding a slowly varying DC bias between the tip and the bottom electrode. Before the measurements, the films were carefully cleaned with ethanol to remove traces of the NaCl/KCl flux.

The relative dielectric permittivity ($\varepsilon_r$) and the dissipation factor (tan $\delta$) were measured in the $10^{-2}$–$10^5$ Hz frequency range using an Alpha-A impedance analyser (Novocontrol Technology). The measurements were performed at a zero bias and 1 V oscillation level. The top electrode (diameter 0.4 mm) was made by sputtering Au through a Kevlar fibre mask on the top of films.

# 3. Results

The deposited thin films were homogeneous with constant thickness, and no cracks or defects were observed by eye. Hence a successful aqueous route to KNN thin films was developed where the amount of organic additives was kept at a minimum.

The $\theta - 2\theta$ XRD patterns of the KNN films deposited using excess alkali on the three different orientations of STO heat treated at 800°C are shown in figure 1a−c. Only (100), (110) and (111) diffraction reflections are observed, demonstrating that the films have conformed the orientation of the substrates. The diffractograms for the films prepared using a salt flux in addition to the excess alkali show similar features (figure 1d−f) to the films prepared without the salt flux. However, grazing incidence diffractograms included in figure 1g,h reveal that the surface of the films also contains a

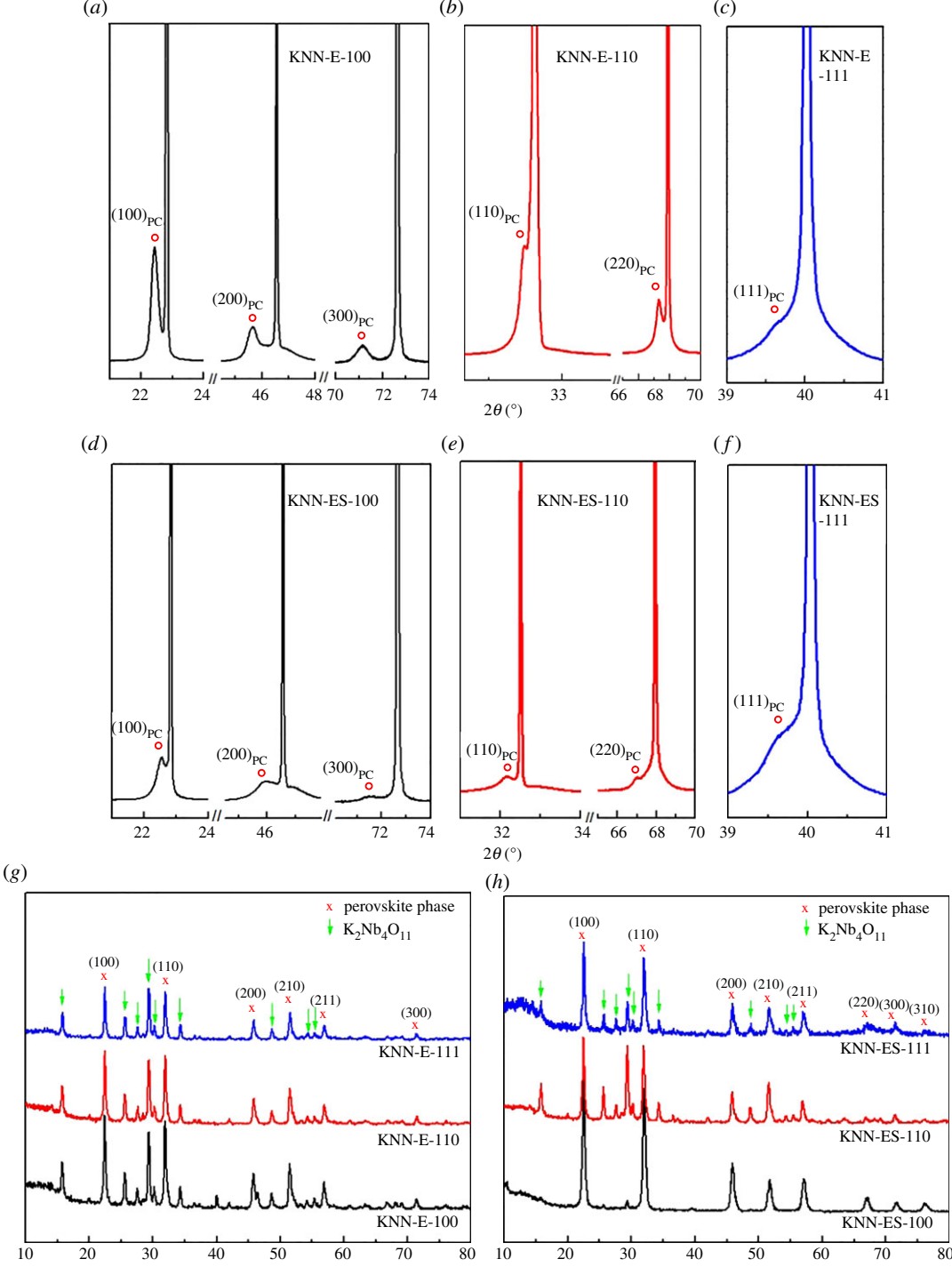

**Figure 1.** ($a$–$f$) Regular $\theta - 2\theta$ diffractograms of KNN-based thin films deposited on STO substrates with different orientations and ($g$,$h$) grazing incidence diffractograms of the same films. The KNN reflections are indexed with pseudocubic indices.

$K_2Nb_4O_{11}$ secondary phase. The use of the salt flux significantly reduced the amount of $K_2Nb_4O_{11}$ formed. The diffraction lines from the secondary phase are not observed in the $\theta - 2\theta$ patterns of the films except for some weak reflections for the film on (111) STO synthetized without salt flux (data not shown), which demonstrates that the secondary phase is mainly present at the surface of the films.

The film thickness and microstructure were studied by TEM, which confirmed that the orientation of the films followed the orientation of the substrates as shown by XRD. Cross-section TEM image of the KNN-ES-100 film is provided in figure 2a, revealing a film thickness in the range 310–345 nm. Close

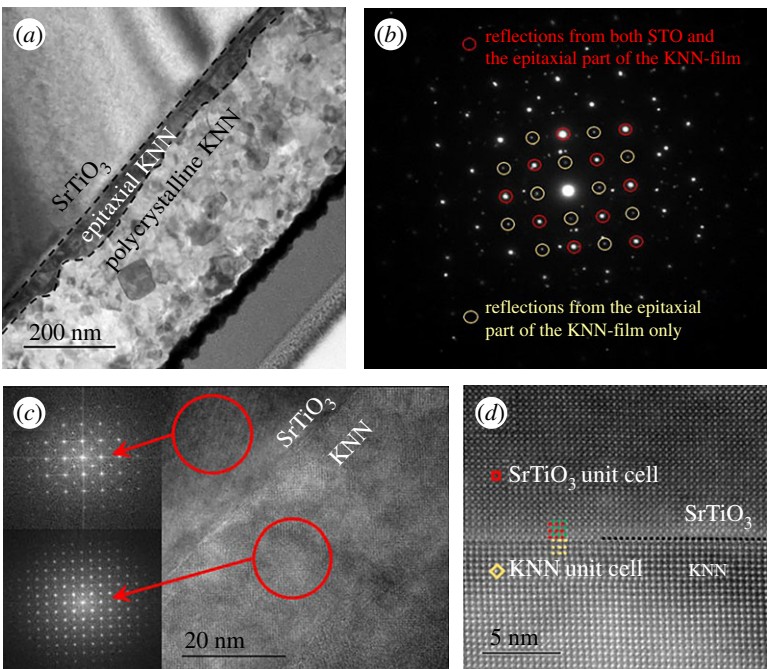

**Figure 2.** (a) TEM image of cross-section of KNN thin film (KNN-ES-100) on (100) STO, thickness 345 nm, (b) electron diffraction pattern of the epitaxial part of the KNN film and the STO substrate (reflections from the epitaxial part of the film are marked with yellow while reflections from both STO and the epitaxial film are marked with red circles), (c) high resolution TEM image of the KNN film substrate interface region with Fourier transforms from the STO substrate and the epitaxial KNN film, and (d) high-angle annular dark field scanning TEM image of the film, including the STO–KNN interface (Nb is marked with yellow, Sr with red and Ti with green dots).

to the STO substrate, an epitaxial film with the thickness of 34–68 nm is observed followed by a dense polycrystalline top layer. The grain size ranges from a few nm up to more than 100 nm. The electron diffraction pattern of the substrate/epitaxial layer in figure 2b depicts that the epitaxial part of the KNN film has extra reflections not present in the substrate (marked by yellow rings) implying that the KNN unit cell is doubled along the $\langle 011 \rangle$ directions. Hence, the unit cell parameters of the epitaxial KNN film are equal to $\sqrt{2} \cdot a_{STO}$ ($a_{STO}$ is the cubic lattice parameter of $SrTiO_3$) and oriented parallel to the $\langle 011 \rangle_{STO}$ directions. The high resolution TEM (HRTEM) image and Fourier transforms from the STO substrate and the epitaxial KNN film given in figure 2c clearly show that the KNN film has a small distortion of the principal cubic perovskite unit cell with a doubling of the unit cell along the cubic $\langle 011 \rangle$ directions. The high-angle annular dark field scanning transmission electron microscopy (HAADF STEM) image (figure 2d) shows that the substrate is SrO terminated and the KNN film starts with an $NbO_2$ plane. The black dotted line marks the atomically sharp interface between the substrate and the film. The STO and KNN unit cells are marked with red and yellow frames, respectively. SEM confirmed the presence of plate-like grains of the $K_2Nb_4O_{11}$ secondary phase covering parts of the surface of the films (data not shown). No secondary phases were detected in the polycrystalline part of the film, confirming the presence of $K_2Nb_4O_{11}$ at the surface inferred from XRD.

Figure 3a presents the TEM image of the cross-section of KNN thin film (KNN-ES-110) on (110) STO. An epitaxial layer is also here observed close to the substrate with a polycrystalline film on top. The epitaxial layer on this substrate shows a more irregular thickness compared to the film on the (100) substrate as growth into pyramids is observed. The polycrystalline film contained some porosity. The diffraction pattern covering the region that includes both the STO substrate, and the epitaxial part of the KNN film is indexed based on an orthorhombic unit cell where $a_o \approx c_o \approx \sqrt{2} \cdot a_{STO}$. In the diffraction pattern of the polycrystalline part of the KNN film only, three red rings are included in the pattern. These rings correspond to the d-spacing of the (100), (110) and (200) reflections (cubic structure) while the diffraction spots between correspond to unique orthorhombic reflections, illustrated with blue rings (not present for the cubic unit cell). The HRTEM image of the KNN film substrate interface region (figure 3b) confirms the epitaxial growth.

The films deposited on (111) STO are epitaxial, and single crystal behaviour is observed all over the film which is oriented with the [111] STO direction perpendicular to the STO/KNN interface. The

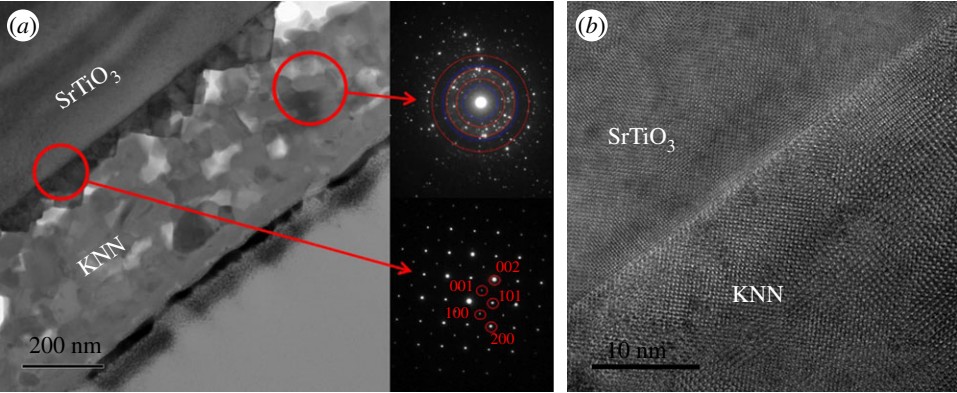

**Figure 3.** (*a*) TEM image of cross-section of KNN thin film (KNN-ES-110) on (110) STO and corresponding electron diffraction patterns of the polycrystalline layer and the STO/epitaxial layer interface region and (*b*) HRTEM image of the KNN film substrate interface region.

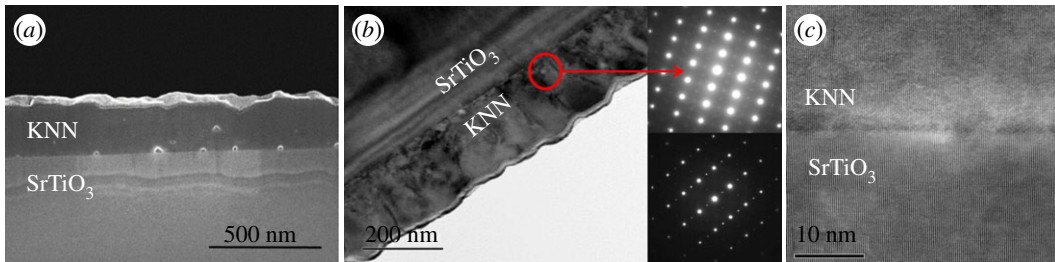

**Figure 4.** (*a*) SE SEM image of cross-section of KNN thin film (KNN-ES-111) on (111) STO, (*b*) TEM image of cross-section and corresponding electron diffraction patterns of two domains in the epitaxial film (the weak reflections in the lower diffraction pattern are accounted for by having $b_o \approx 2a_{pc}$) and (*c*) HRTEM image of the KNN film substrate interface region.

secondary electron SEM (SE SEM) image of a cross-section of the film in figure 4*a* shows a dense film with only a few pores located at the STO/KNN interface. The TEM image presented in figure 4*b* confirms a dense film with a thickness in the range from 180 to 220 nm. The electron diffraction patterns of the film mostly show domains with $b_o \approx a_{pc}$ ($a_{pc}$ is the pseudocubic lattice parameter), but also some domains with $b_o \approx 2 \cdot a_{pc}$ are observed as shown in the lower pattern. The HRTEM image of KNN-ES-111 in figure 4*c* shows the epitaxy relation between the film and substrate.

The PR amplitude and phase, measured by PFM of the thin films at room temperature, are shown in figure 5. The PR amplitudes exhibit well-shaped butterfly loops and the observed change in the PR phase with reversal of the polarization confirms that the films are ferroelectric. The PR amplitude varies little between substrate orientations studied. The PR phase hysteresis loop investigations show a saturation phase switch at high applied field, and the PR phase shift is in the range of $180°$ in accordance with small amounts of secondary phase in these films.

The dielectric properties of the KNN-ES-100 and KNN-ES-110 thin films are presented in figure 6. The higher losses at low frequencies might be due to the adsorption of moisture on the film surface. The permittivity and loss tangent of KNN-ES-100 are 870 and 0.04 and for KNN-ES-110 2250 and 0.025, respectively, at 1 kHz.

# 4. Discussion

The XRD patterns of the KNN-based films on STO substrates (figure 1) demonstrated a strong preferential orientation following the crystallographic orientation of the substrate as only the (h00), (0k0) and (00l) diffraction lines are present respectively on the three different orientations of the substrate (except for some weak lines of (100) and (110) on the (111) substrate). The lower $2\theta$ values observed for the diffraction lines of the films, as compared with the substrates, reveal that the films have a larger out-of-plane lattice parameter than the substrate (figure 1*a*–*f*). An important feature of thin films deposited on a substrate is the presence of two-dimensional in-plane strain [19]. The origin of the misfit strain includes the difference in lattice parameters and the thermal expansion due to

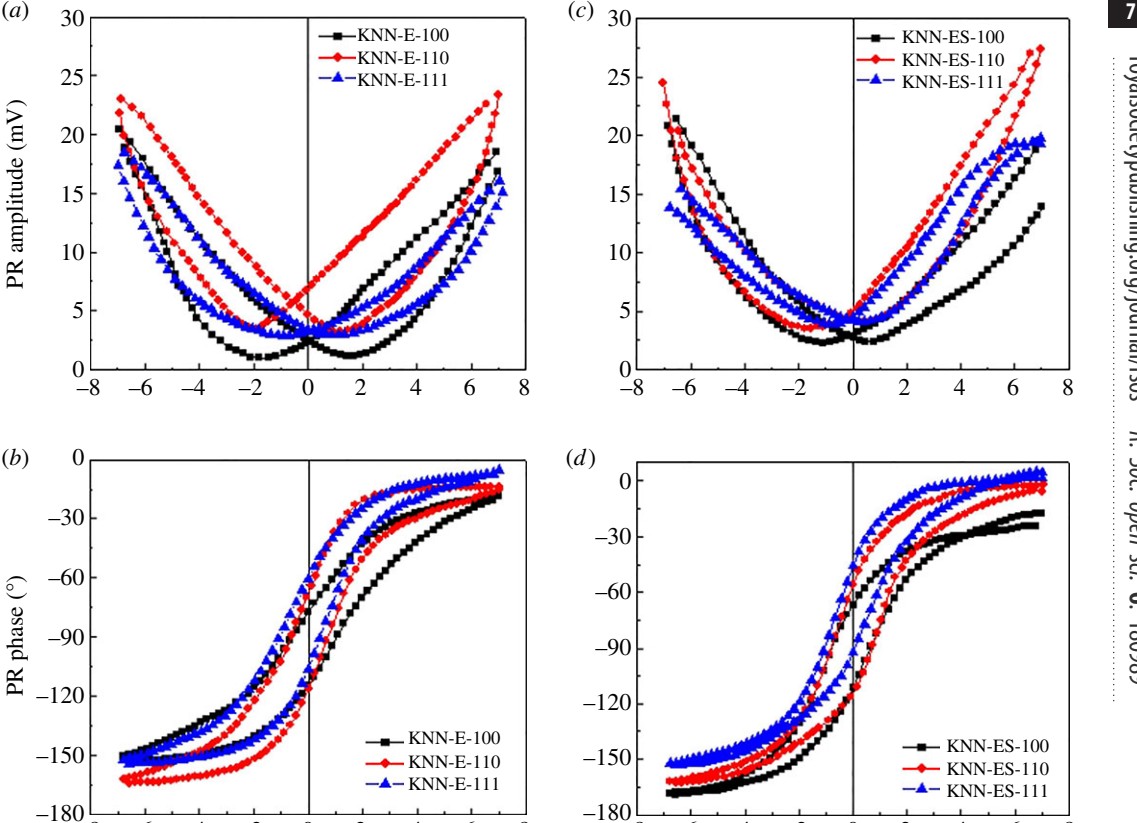

**Figure 5.** (a) Piezoresponse amplitude and (b) PR phase of KNN thin films on (100), (110) and (111) Nb:STO substrates from the solution with alkali excess. (c,d) Corresponding measurements on films from the solution with alkali excess and NaCl/KCl flux.

different thermal expansion coefficients of the film and the substrate [20]. The thermal expansion of KNN is lower than for STO [21,22]. After heat treatment at 800°C, the KNN film will therefore be in compression.

Epitaxial growth is most usually obtained by physical deposition methods [6–8], but epitaxial growth was clearly obtained by the simple aqueous chemical solution deposition method developed in this work (figure 4). Similar films to the ones obtained here have also been prepared by sol–gel deposition of KNN films using alkoxides and an organic solvent [23]. Yu *et al.* [23] proposed that the development of the epitaxial layer was dependent on the pyrolysis temperature and observed a higher probability for epitaxy when the pyrolysis temperature was in the range 400–450°C. This is in accordance with our work where a pyrolysis temperature of 400°C was used. A low pyrolysis temperature will induce heterogeneous nucleation to occur at the surface of the substrates due to the lower activation energy for heterogeneous compared with homogeneous nucleation [24]. Even though it is promising to fabricate epitaxial KNN films by this cheap and simple method, more studies with respect to processing parameters are necessary to elucidate the mechanism for the epitaxial growth. For the films on (100) and (110) oriented STO, which contained an epitaxial layer and a polycrystalline part, both heterogeneous nucleation took place promoting the epitaxy and homogeneous nucleation resulting in the polycrystalline part of the film. In the epitaxial films on (111) STO substrate, a few pores were observed close to the film substrate interface. Yu *et al.* [23] claimed that the large porosity in their epitaxial films was caused by decomposition of alkali carbonates formed by the combustion of organics, forming $CO_2$. The uniform KNN films obtained by the method presented here might be explained by the advantage of using inorganic compared to organic precursors resulting in less effective volatility. During the drying of the films, the use of water might be important to facilitate shrinkage due to the high surface tension of water compared to solvents like 2-methoxyethanol as long as the film thickness is small.

The GIXRD patterns and TEM of the KNN thin films confirmed the presence of the $K_2Nb_4O_{11}$ secondary phase at the surface of the films. The formation of the niobium-rich phase can be explained by the high volatility of K and Na species during the heat treatment. The amount of secondary phase

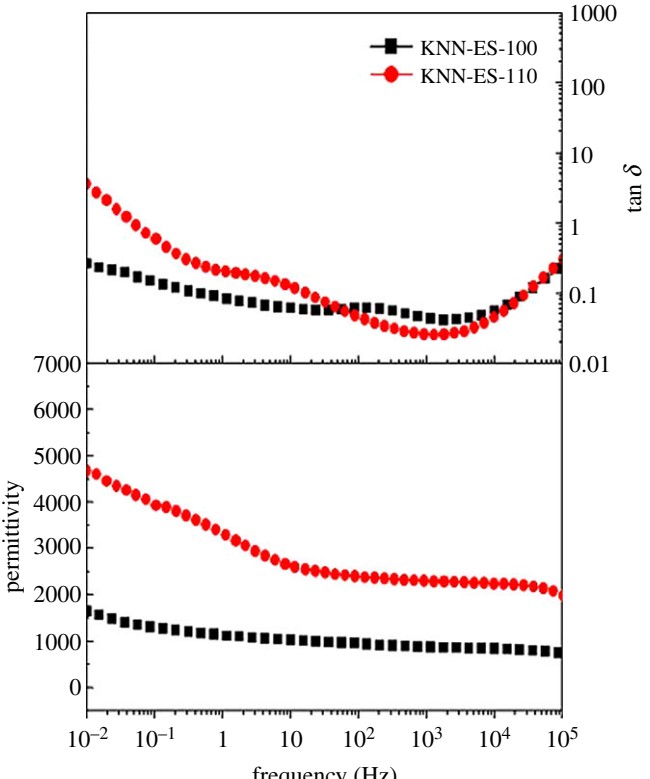

**Figure 6.** Dielectric permittivity and tan $\delta$ of KNN-based films (KNN-ES-100) on (100) STO and (KNN-ES-110) on (110) STO heat treated at 800°C.

caused by the loss of alkali species was independent of heat treatment temperature in the range 700–900°C (data not shown). Hence, direct evaporation of K(g) or Na(g) from KNN according to equation (4.1) (simplified for $KNbO_3$) cannot explain the formation of the secondary phases observed from these films.

$$4KNbO_3(s) = 2K(g) + \tfrac{1}{2}O_2(g) + K_2Nb_4O_{11}(s). \tag{4.1}$$

The vapour pressure of K(g) over $KNbO_3$ starts to be significant for loss of alkali at rather high temperatures [25] and should not be significant at the lowest heat treatment temperatures used here. Another mechanism for the loss of alkali should therefore be considered and the formation of alkali hydroxide/carbonate phases at a lower temperature according to equations (4.2) and (4.3) (simplified reactions with only K) has previously been proposed [25]. Even if the films in this work were heat treated in pure oxygen, the films were exposed to ambient air and the decomposition of the oxalate and small amount of PVA added as a wetting agent could form $CO_2$.

$$K_2O(s) + CO_2(g) = K_2CO_3(s) \tag{4.2}$$

and

$$K_2O(s) + H_2O(g) = 2KOH(s). \tag{4.3}$$

Volatilization of these alkali hydroxide/carbonate compounds which form a melt at about 650°C [25] according to equations (4.4) and (4.5) is therefore the most plausible reason for the alkali loss of the films and the formation of the niobium-rich $K_2Nb_4O_{11}$ secondary phase.

$$K_2CO_3(l,s) = 2K(g) + \tfrac{1}{2}O_2(g) + CO_2(g) \tag{4.4}$$

and

$$2KOH(l,s) = 2K(g) + \tfrac{1}{2}O_2(g) + H_2O(g). \tag{4.5}$$

The presence of the NaCl/KCl flux during the processing reduces the amount of $K_2Nb_4O_{11}$ significantly (figure 1). According to the NaCl–KCl phase diagram [26], the KCl–NaCl eutectic has a melting point of 650°C. During the heat treatment, we propose that the alkali hydroxides/carbonates formed are dissolved in the salt flux due to the ionic character of the species. In order to explain the

effect of the salt on reducing the $K_2Nb_4O_{11}$ content, we deduce that the ionic liquid reduces the activity of the alkalis, hence reducing the volatilization.

The PR phase data reported in figure 5 clearly show that the films are ferroelectric. As no significant effect of the substrate orientation on the piezo properties was observed, we propose that due to the compressive stress in the films, the polarization direction is aligned vertically in all the films giving a similar response. The dielectric properties of selected films are illustrated in figure 6. The measured dielectric constant of the films was shown to be higher and the loss tangent lower than most literature data reported for films prepared by CSD and sol−gel methods [16−18,27−29].

# 5. Conclusion

Homogeneous KNN thin films with a thickness of a few hundred nm were prepared by aqueous chemical solution deposition followed by heat treatment at 800°C. The high degree of orientation, high local piezoelectric properties and high permittivity were evidenced by XRD, PFM and dielectric spectroscopy, respectively. The amount of secondary phase was reduced by the use of alkali excess and a salt flux during the processing. The high degree of orientation of the films demonstrates the advantages of our environmentally friendly synthesis method developed, which can be applicable for the fabrication of oriented and epitaxial lead-free ferroelectric thin films.

Ethics. Our study does not use humans or human tissue for data collection and does not need a research ethics approval.
Data accessibility. The XRD data, thermogravimetry data, piezoresponse phase and amplitude data, permittivity data and transmission electron image are available: http://dx.doi.org/10.5061/dryad.g05350g [30].
Authors' contributions. K.-N.P., T.G. and M.-A.E. conceived and designed the experiments; P.E.V. performed the TEM experiments; K.-N.P. and M.M. performed the electrical measurements; K.-N.P performed the synthesis experiments; K.-N.P., T.T., T.G. and M.-A.E. analysed the data; K.-N.P., N.H.G. and M.-A.E. wrote the paper with inputs from all the authors. All authors gave final approval for publication.
Competing interests. We have no competing interests.
Funding. The Research Council of Norway is acknowledged for the support to the FRINATEK LEADFREE project no. 197497/F20, NANO2021 project PIEZOMED project no. 250184 and Norwegian Micro- and Nano-Fabrication Facility, NorFab, project no. 245963/F50. The TEM work was carried out on NORTEM infrastructure, Grant 197405, TEM Gemini Centre, Norwegian University of Science and Technology (NTNU), Norway.

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
