## [Reviewer comments · Royal Society Open Science]

Review History

RSOS-180989.R0 (Original submission)

Review form: Reviewer 1

Is the manuscript scientifically sound in its present form?

No

Are the interpretations and conclusions justified by the results?

No

Is the language acceptable?

Yes

Is it clear how to access all supporting data?

No

Do you have any ethical concerns with this paper?

No

Have you any concerns about statistical analyses in this paper?

I do not feel qualified to assess the statistics

Recommendation?

Major revision is needed (please make suggestions in comments)

Comments to the Author(s)

The manuscript on aqueous CSD of epitaxial KNN thin films brings new results, but it opens more questions than it provides answers.

1. Use and role of KCl/NaCl flux

Regarding the use of KCl/NaCl flux, it is written in Experimental part that a KCl/NaCl (25 mol%) was added to selected precursor solutions... Please explain with more detail how the precursor solution with KCl/NaCl flux was prepared, and what the exact formulation was. If the chlorides were added to the aqueous solution, they should be dissolved, consequently the excess of alkalines would be increased. Chloride ions should also be present in the solution, residues could remain in the film. Many scenarios are possible without knowing the exact procedure. Are there any results that support the formation and the role of the KCl/NaCl flux upon RTA as proposed in the Discussion?

Equations 2 – 5 are possible, but they would require at least some confirmation. Could any surface sensitive technique help to detect the carbonate groups? Was anything tried? The authors wrote that these reactions were proposed in ref. 25, but they were not confirmed neither in the mentioned reference, nor in the present manuscript.

Does the presence of flux influence the thickness of the films? The thicknesses of E-films are not reported; is there any relation between E and ES groups of the films?

The Results include the XRD patterns of both groups of films (E, ES), but the TEM study was performed only on the ES films. Was there any difference between the two groups – thickness, microstructure, epitaxy (if evident), porosity distribution, etc. Please comment.

2. Thermal expansion of KNN

Phase transitions of KNN Cubic – tetragonal – orthorhombic are manifested by changes in cell parameters, and also in thermal expansion, similar as in other ferroelectrics. So, please clarify the statement that thermal expansion of KNN is almost independent on temperature. The studied KNN films exhibit local ferroelectric response, so they should undergo a sequence of phase transitions upon cooling.

3. Heterogeneous vs. Homogeneous nucleation

In the 2nd paragraph of Discussion the authors provide explanation for the epitaxial growth of KNN films. They relate the low pyrolysis temperature (400 °C) with the induced heterogeneous nucleation. So, if the pyrolysis temperature was higher was this not the case? 400 °C is not an exceptionally low temperature for pyrolysis. Furthermore the reference they are referring to (24, Schwartz and Schneller) discusses PZT and (Ba,Sr)TiO₃ thin films prepared from organic precursors, and low / high temperature in such materials may not be directly transferred to water-based CSD of KNN films.

The authors further write that in (100) and (110) films there is a competition between hetero- and homogeneous nucleation. Is then the growth of the nucleated phase a problem? Is there any difference in thicknesses of epi- and polycrystalline parts of respective films if the annealing conditions are changed?

4. Secondary K₂Nb₄O₁₁ (2,4-) phase

The authors write that the 2,4 -phase could only be seen by GIXRD. Does this mean that e.g. log scale or extended step time of conventional XRD patterns revealed nothing? The GIXRD patterns of individual samples still differ quite a lot.

The secondary 2,4-phase is discussed in Discussion, 3rd paragraph. It is written that the mass loss was significant and that direct evaporation of alkalis can not explain such mass loss. Until this paragraph mass loss has not been even mentioned. Please explain, what this mass loss means and how it has been determined (if at all – it is a challenging task to do it in a film). How is it confirmed that 'direct evaporation of K and Na from KNN ... can not explain the mass loss observed from these films.' Please explain and provide evidence.

The majority of alkali polyniobate phases is humidity sensitive. Was there any sign of humidity sensitivity in the studied films? For example was there any difference in dielectric properties if they were measured in vacuum or in air (this refers also to the comment no. 5)?

Was the 2,4 phase detected in any of the samples by SEM/TEM? A comment on detection of this phase is definitely missing.

5. PFM, dielectric properties

PFM results are shown for both E and ES groups revealing that all films show local ferroelectric response. However, macroscopic measurements are reported only for two of the ES films, so one could assume that all other films were too leaky. Very high losses at low frequency could be presumably related to interaction of the films with humidity.

Are different permittivities of KNN (100) and (110) films related to orientational dependence?

Based on the results of dielectric measurements it is difficult to accept the last phrase of the Abstract, namely that 'The films demonstrated relatively high permittivity with low loss. This phrase is much too general and even misleading, it should be modified to reflect the results more precisely.

Technical comments:

The formula of KNN in Summary, 2nd line, should be written as in the rest of the manuscript: $K_{0.5}Na_{0.5}NbO_3$.

Ref. 19 – the 3rd author is Tagantsev.

Review form: Reviewer 2

Is the manuscript scientifically sound in its present form?

Yes

Are the interpretations and conclusions justified by the results?

Yes

Is the language acceptable?

Yes

Is it clear how to access all supporting data?

Not Applicable

Do you have any ethical concerns with this paper?

No

Have you any concerns about statistical analyses in this paper?

No

Recommendation?

Accept with minor revision (please list in comments)

Comments to the Author(s)

- 1) Epitaxial structured KNN thin films were obtained using a chemical method. A change from an epitaxial to polycrystalline structure was observed in some thin films.
- 2) The plate-like grains of the secondary phase $K_2Nb_4O_{11}$ were confirmed on the surface of the thin films.
- 3) There are some acronyms undefined, or used for only 1 time, e.g., MOCVD, FEI, EELS, TEM, HRTEM, SESEM.

Decision letter (RSOS-180989.R0)

02-Oct-2018

Dear Professor Einarsrud:

Title: Epitaxial $K_{0.5}Na_{0.5}NbO_3$ thin films by aqueous chemical solution deposition
Manuscript ID: RSOS-180989

The editor assigned to your manuscript has now received comments from reviewers. We would like you to revise your paper in accordance with the referee and Subject Editor suggestions which can be found below (not including confidential reports to the Editor). Please note this decision does not guarantee eventual acceptance.

Please submit your revised paper before 25-Oct-2018. Please note that the revision deadline will expire at 00.00am on this date. If we do not hear from you within this time then it will be assumed that the paper has been withdrawn. In exceptional circumstances, extensions may be possible if agreed with the Editorial Office in advance. We do not allow multiple rounds of revision so we urge you to make every effort to fully address all of the comments at this stage. If deemed necessary by the Editors, your manuscript will be sent back to one or more of the original reviewers for assessment. If the original reviewers are not available we may invite new reviewers.

Once again, thank you for submitting your manuscript to Royal Society Open Science and I look

forward to receiving your revision. If you have any questions at all, please do not hesitate to get in touch.

Yours sincerely,
 Dr Laura Smith, MRSC
 Publishing Editor, Journals
 Royal Society of Chemistry,
 Thomas Graham House,
 Science Park, Milton Road,
 Cambridge, CB4 0WF, UK

Royal Society Open Science - Chemistry Editorial Office

On behalf of the Subject Editor Professor Anthony Stace and the Associate Editor Professor Claire Carmalt.

RSC Associate Editor:
 Comments to the Author:
 (There are no comments.)

RSC Subject Editor:
 Comments to the Author:
 (There are no comments.)

Reviewers' Comments to Author:
 Reviewer: 1

Comments to the Author(s)

The manuscript on aqueous CSD of epitaxial KNN thin films brings new results, but it opens more questions than it provides answers.

1. Use and role of KCl/NaCl flux

Regarding the use of KCl/NaCl flux, it is written in Experimental part that a KCl/NaCl (25 mol%) was added to selected precursor solutions... Please explain with more detail how the precursor solution with KCl/NaCl flux was prepared, and what the exact formulation was. If the chlorides were added to the aqueous solution, they should be dissolved, consequently the excess of alkalines would be increased. Chloride ions should also be present in the solution, residues could remain in the film. Many scenarios are possible without knowing the exact procedure. Are there any results that support the formation and the role of the KCl/NaCl flux upon RTA as proposed in the Discussion?

Equations 2 - 5 are possible, but they would require at least some confirmation. Could any surface sensitive technique help to detect the carbonate groups? Was anything tried? The authors wrote that these reactions were proposed in ref. 25, but they were not confirmed neither in the mentioned reference, nor in the present manuscript.

Does the presence of flux influence the thickness of the films? The thicknesses of E-films are not reported; is there any relation between E and ES groups of the films?

The Results include the XRD patterns of both groups of films (E, ES), but the TEM study was performed only on the ES films. Was there any difference between the two groups - thickness, microstructure, epitaxy (if evident), porosity distribution, etc. Please comment.

2. Thermal expansion of KNN

Phase transitions of KNN Cubic – tetragonal – orthorhombic are manifested by changes in cell parameters, and also in thermal expansion, similar as in other ferroelectrics. So, please clarify the statement that thermal expansion of KNN is almost independent on temperature. The studied KNN films exhibit local ferroelectric response, so they should undergo a sequence of phase transitions upon cooling.

3. Heterogeneous vs. Homogeneous nucleation

In the 2nd paragraph of Discussion the authors provide explanation for the epitaxial growth of KNN films. They relate the low pyrolysis temperature (400 °C) with the induced heterogeneous nucleation. So, if the pyrolysis temperature was higher was this not the case? 400 °C is not an exceptionally low temperature for pyrolysis. Furthermore the reference they are referring to (24, Schwartz and Schneller) discusses PZT and (Ba,Sr)TiO₃ thin films prepared from organic precursors, and low / high temperature in such materials may not be directly transferred to water-based CSD of KNN films.

The authors further write that in (100) and (110) films there is a competition between hetero- and homogeneous nucleation. Is then the growth of the nucleated phase a problem? Is there any difference in thicknesses of epi- and polycrystalline parts of respective films if the annealing conditions are changed?

4. Secondary K₂Nb₄O₁₁ (2,4-) phase

The authors write that the 2,4 -phase could only be seen by GIXRD. Does this mean that e.g. log scale or extended step time of conventional XRD patterns revealed nothing? The GIXRD patterns of individual samples still differ quite a lot.

The secondary 2,4-phase is discussed in Discussion, 3rd paragraph. It is written that the mass loss was significant and that direct evaporation of alkalis can not explain such mass loss. Until this paragraph mass loss has not been even mentioned. Please explain, what this mass loss means and how it has been determined (if at all – it is a challenging task to do it in a film). How is it confirmed that 'direct evaporation of K and Na from KNN ... can not explain the mass loss observed from these films.' Please explain and provide evidence.

The majority of alkali polyniobate phases is humidity sensitive. Was there any sign of humidity sensitivity in the studied films? For example was there any difference in dielectric properties if they were measured in vacuum or in air (this refers also to the comment no. 5)?

Was the 2,4 phase detected in any of the samples by SEM/TEM? A comment on detection of this phase is definitely missing.

5. PFM, dielectric properties

PFM results are shown for both E and ES groups revealing that all films show local ferroelectric response. However, macroscopic measurements are reported only for two of the ES films, so one could assume that all other films were too leaky. Very high losses at low frequency could be presumably related to interaction of the films with humidity.

Are different permittivities of KNN (100) and (110) films related to orientational dependence?

Based on the results of dielectric measurements it is difficult to accept the last phrase of the Abstract, namely that 'The films demonstrated relatively high permittivity with low loss. This phrase is much too general and even misleading, it should be modified to reflect the results more precisely.

Technical comments:

The formula of KNN in Summary, 2nd line, should be written as in the rest of the manuscript: K_{0.5}Na_{0.5}NbO₃.

Ref. 19 – the 3rd author is Tagantsev.

Reviewer: 2

Comments to the Author(s)

- 1) Epitaxial structured KNN thin films were obtained using a chemical method. A change from an epitaxial to polycrystalline structure was observed in some thin films.
- 2) The plate-like grains of the secondary phase $K_2Nb_4O_{11}$ were confirmed on the surface of the thin films.
- 3) There are some acronyms undefined, or used for only 1 time, e.g., MOCVD, FEI, EELS, TEM, HRTEM, SESEM.

Author's Response to Decision Letter for (RSOS-180989.R0)

See Appendix A.

RSOS-180989.R1 (Revision)

Review form: Reviewer 1

Is the manuscript scientifically sound in its present form?

Yes

Are the interpretations and conclusions justified by the results?

Yes

Is the language acceptable?

Yes

Is it clear how to access all supporting data?

Yes

Do you have any ethical concerns with this paper?

No

Have you any concerns about statistical analyses in this paper?

I do not feel qualified to assess the statistics

Recommendation?

Accept as is

Comments to the Author(s)

The authors have addressed the comments of the reviewer with explanation and clarifications wherever needed. Appropriate adjustments have been made in the revised version of the manuscript.

Decision letter (RSOS-180989.R1)

13-Nov-2018

Dear Professor Einarsrud:

Title: Epitaxial $K_{0.5}Na_{0.5}NbO_3$ thin films by aqueous chemical solution deposition
Manuscript ID: RSOS-180989.R1

It is a pleasure to accept your manuscript in its current form for publication in Royal Society Open Science. The chemistry content of Royal Society Open Science is published in collaboration with the Royal Society of Chemistry.

On behalf of the Subject Editor Professor Anthony Stace and the Associate Editor Professor Claire Carmalt.

RSC Associate Editor:
Comments to the Author:
(There are no comments.)

RSC Subject Editor:
Comments to the Author:
(There are no comments.)

Reviewer(s)' Comments to Author:
Reviewer: 1

Comments to the Author(s)
The authors have addressed the comments of the reviewer with explanation and clarifications wherever needed. Appropriate adjustments have been made in the revised version of the manuscript.

Appendix A

Response to referees

Epitaxial $K_{0.5}Na_{0.5}NbO_3$ thin films by aqueous chemical solution deposition

Manuscript ID: RSOS-180989

The authors are grateful to the comments from the referees which are helpful in improving the quality of our manuscript. Please find below our response to the comments from the referees.

Reviewers' Comments to Author:

Reviewer: 1

Comments to the Author(s)

The manuscript on aqueous CSD of epitaxial KNN thin films brings new results, but it opens more questions than it provides answers.

1. Use and role of KCl/NaCl flux

Regarding the use of KCl/NaCl flux, it is written in Experimental part that a KCl/NaCl (25 mol%) was added to selected precursor solutions... Please explain with more detail how the precursor solution with KCl/NaCl flux was prepared, and what the exact formulation was. If the chlorides were added to the aqueous solution, they should be dissolved, consequently the excess of alkalines would be increased. Chloride ions should also be present in the solution, residues could remain in the film. Many scenarios are possible without knowing the exact procedure.

Are there any results that support the formation and the role of the KCl/NaCl flux upon RTA as proposed in the Discussion?

Equations 2 – 5 are possible, but they would require at least some confirmation. Could any surface sensitive technique help to detect the carbonate groups? Was anything tried? The authors wrote that these reactions were proposed in ref. 25, but they were not confirmed neither in the mentioned reference, nor in the present manuscript.

Does the presence of flux influence the thickness of the films? The thicknesses of E-films are not reported; is there any relation between E and ES groups of the films?

The Results include the XRD patterns of both groups of films (E, ES), but the TEM study was performed only on the ES films. Was there any difference between the two groups – thickness, microstructure, epitaxy (if evident), porosity distribution, etc. Please comment.

Authors comment:

The authors are grateful to the referee for several very interesting comments here. A more detailed description about how the salts were added are included in the revised manuscript. In addition, the amount of chlorides added is clarified. The chlorides were dissolved in the aqueous precursor solution and correctly the concentration of alkali in the solution increased. However, since 1:1 molar ratio mixture of NaCl/KCl has a melting temperature of 650 °C the chloride mixture will form a molten flux during the heat treatment of the films. The reason for a lower amount of the secondary phase in the films with the salt flux is therefore proposed to be the reduced evaporation of volatile alkali species dissolved in the flux and not the fact that there is a higher total alkali concentration in the precursor solution. At the final temperature of thermal treatment of the films, the chlorides have a significant vapor pressure. No chlorides were observed in the films during the TEM studies. Unfortunately we were not able to directly observe the flux formation during the heat treatment, the only indirect observation we have is the reduced amount of secondary phases in the films with the salt added to the precursor solution.

With respect to the formation of carbonates/hydroxides of the alkali metals we have previously detected carbonate species in bulk KNN samples by FTIR spectroscopy confirming our hypothesis (unpublished data).

Unfortunately we have only been able to study the ES-series films by TEM due to budget reasons. Since the ES-series of films contained less secondary phases we focused the present TEM study on them. We therefore have no data on the thickness of the E-series of films and also limited other data.

2. Thermal expansion of KNN

Phase transitions of KNN Cubic – tetragonal – orthorhombic are manifested by changes in cell parameters, and also in thermal expansion, similar as in other ferroelectrics. So, please clarify the statement that thermal expansion of KNN is almost independent on temperature. The studied KNN films exhibit local ferroelectric response, so they should undergo a sequence of phase transitions upon cooling.

Authors comments:

The authors are grateful for this comment. There are variations in the lattice parameter of KNN due to the phase transitions as described by the reviewer. The most important point however is that the thermal expansion coefficient for KNN is smaller than for the SrTiO₃ substrate and the film will then during cooling be forced into compression. The revised manuscript has been modified by removing the sentence about the “almost constant thermal

expansion” as this is obviously wrong. More relevant references have been included.

3. Heterogeneous vs. Homogeneous nucleation

In the 2nd paragraph of Discussion the authors provide explanation for the epitaxial growth of KNN films. They relate the low pyrolysis temperature (400 oC) with the induced heterogeneous nucleation. So, if the pyrolysis temperature was higher was this not the case? 400 oC is not an exceptionally low temperature for pyrolysis. Furthermore the reference they are referring to (24, Schwartz and Schneller) discusses PZT and (Ba,Sr)TiO₃ thin films prepared from organic precursors, and low / high temperature in such materials may not be directly transferred to water-based CSD of KNN films. The authors further write that in (100) and (110) films there is a competition between hetero- and homogeneous nucleation. Is then the growth of the nucleated phase a problem? Is there any difference in thicknesses of epi- and polycrystalline parts of respective films if the annealing conditions are changed?

Authors comment:

Thanks to the referee for this comment which shows that we need to clarify the text in the 2nd paragraph of the Discussion. The low and high pyrolysis temperatures described are meant as relative values where low temperatures are facilitating heterogeneous nucleation (REF Schwartz and Schneller) due to lower activation energy and higher temperatures are favoring homogeneous nucleation. In that respect it is not important which materials that are discussed in the reference. The text in the 2nd paragraph of the Discussion has been revised to clarify.

Regarding the (100) and (110) substrates we have also modified the text to make it clear that both homogeneous and heterogeneous nucleation is taking place giving the two different types of microstructures. We do unfortunately not have any information about how the thickness of the epitaxial and polycrystalline layers are changing with other processing parameters.

4. Secondary K₂Nb₄O₁₁ (2,4-) phase

The authors write that the 2,4 –phase could only be seen by GIXRD. Does this mean that e.g. log scale or extended step time of conventional XRD patterns revealed nothing? The GIXRD patterns of individual samples still differ quite a lot.

The secondary 2,4-phase is discussed in Discussion, 3rd paragraph. It is written that the mass loss was significant and that direct evaporation of alkalis can not explain such mass loss. Until this paragraph mass loss has not been even mentioned. Please explain, what this mass loss means and how it has been determined (if at all – it is a challenging task to do it in a film). How

is it confirmed that ' direct evaporation of K and Na from KNN ... can not explain the mass loss observed from these films.' Please explain and provide evidence.

The majority of alkali polyniobate phases is humidity sensitive. Was there any sign of humidity sensitivity in the studied films? For example was there any difference in dielectric properties if they were measured in vacuum or in air (this refers also to the comment no. 5)?

Was the 2,4 phase detected in any of the samples by SEM/TEM? A comment on detection of this phase is definitely missing.

Authors comment:

The $K_2Nb_4O_{11}$ secondary phase was observed as plate-like grains (few micron in size) spread on the top of the polycrystalline layer by SEM during the FIB preparation of the TEM samples. The secondary phase was also observed as a thin plate-like grain in some of the TEM images of cross-sections as most of the film surface was not covered by these grains. The fact that the secondary phases are located as thin plate-like grains on the surface is the reason why it is more visible in the GIXRD, than in normal XRD. A comment that the secondary phase has been observed in SEM is included in the revised manuscript.

The mass loss of the films was not measured directly, but it is related to the total amount of secondary phase present in the films. This relation is clarified in the revised version of the manuscript by rewriting this part. We are grateful to the referee for pointing out this as we see it can be confusing.

The films were tested for stability by soaking in water and this harsh treatment changed the film surface. However, exposure to ambient air was not observed to change the properties of the films. With our set-up dielectric properties were only measured in ambient air.

5. PFM, dielectric properties

PFM results are shown for both E and ES groups revealing that all films show local ferroelectric response. However, macroscopic measurements are reported only for two of the ES films, so one could assume that all other films were too leaky. Very high losses at low frequency could be presumably related to interaction of the films with humidity.

Are different permittivities of KNN (100) and (110) films related to orientational dependence?

Based on the results of dielectric measurements it is difficult to accept the last phrase of the Abstract, namely that 'The films demonstrated relatively high permittivity with low loss. This phrase is much too general and even misleading, it should be modified to reflect the results more precisely.

Authors comment:

The intension with these measurements was to show that the films are ferroelectric. Unfortunately we measured the macroscopic dielectric properties for only the two films presented and can therefore not provide data for the E-series or any other films. The high losses at low frequency might be due to humidity as proposed by the referee and a sentence about this is included in the revised manuscript. This proposal is supported by the fact that the films changes surface microstructure after a harsh treatment by soaking them in water for 30 min. Since we only have the measurements on the two films, we will not conclude that the different permittivities are related to the different orientation of the substrates. The last sentence of the abstract has been rephrased to: "The permittivity and loss tangent of the films with the salt flux were 870 and 0.04 (100-orientation) and 2250 and 0.025 (110-orientation), respectively, at 1 kHz."

Technical comments:

The formula of KNN in Summary, 2nd line, should be written as in the rest of the manuscript: $\text{K}_{0.5}\text{Na}_{0.5}\text{NbO}_3$.

Ref. 19 – the 3rd author is Tagantsev.

Authors comment:

These two misprints have been corrected.

Reviewer: 2

Comments to the Author(s)

- 1) Epitaxial structured KNN thin films were obtained using a chemical method. A change from an epitaxial to polycrystalline structure was observed in some thin films.
- 2) The plate-like grains of the secondary phase $\text{K}_2\text{Nb}_4\text{O}_{11}$ were confirmed on the surface of the thin films.
- 3) There are some acronyms undefined, or used for only 1 time, e.g., MOCVD, FEI, EELS, TEM, HRTEM, SESEM.

Authors comment:

All acronyms have been defined in the text. FEI is the name of the company.